# Metabolomic Profiling Reveals Distinct and Mutual Effects of Diet and Inflammation in Shaping Systemic Metabolism in *Ldlr^−/−^* Mice

**DOI:** 10.3390/metabo10090336

**Published:** 2020-08-19

**Authors:** Mario A. Lauterbach, Eicke Latz, Anette Christ

**Affiliations:** 1Institute of Innate Immunity, University Hospital Bonn, University of Bonn, 53127 Bonn, Germany; mario.lauterbach@uni-bonn.de; 2Department of Infectious Diseases and Immunology, UMass Medical School, Worcester, MA 01605, USA; 3Center of Molecular Inflammation Research, Norwegian University of Science and Technology, 7491 Trondheim, Norway; 4German Center for Neurodegenerative Diseases (DZNE), 53127 Bonn, Germany

**Keywords:** western-type diets, lipopolysaccharide (LPS), systemic metabolism, systemic inflammation, metabolomic profiling, long-term metabolic rewiring

## Abstract

Changes in modern dietary habits such as consumption of Western-type diets affect physiology on several levels, including metabolism and inflammation. It is currently unclear whether changes in systemic metabolism due to dietary interventions are long-lasting and affect acute inflammatory processes. Here, we investigated how high-fat diet (HFD) feeding altered systemic metabolism and the metabolomic response to inflammatory stimuli. We conducted metabolomic profiling of sera collected from *Ldlr*^−/−^ mice on either regular chow diet (CD) or HFD, and after an additional low-dose lipopolysaccharide (LPS) challenge. HFD feeding, as well as LPS treatment, elicited pronounced metabolic changes. HFD qualitatively altered the systemic metabolic response to LPS; particularly, serum concentrations of fatty acids and their metabolites varied between LPS-challenged mice on HFD or CD, respectively. To investigate whether systemic metabolic changes were sustained long-term, mice fed HFD were shifted back to CD after four weeks (HFD > CD). When shifted back to CD, serum metabolites returned to baseline levels, and so did the response to LPS. Our results imply that systemic metabolism rapidly adapts to dietary changes. The profound systemic metabolic rewiring observed in response to diet might affect immune cell reprogramming and inflammatory responses.

## 1. Introduction

The prevalence of obesity and associated immuno-metabolic diseases has reached pandemic proportions in modern Western societies. Permanent overnutrition of high-calorie convenience products, a sedentary lifestyle, alcohol abuse, stress, lack of sleep and direct exposure to environmental biotoxins, actively contribute to developing non-communicable lifestyle diseases (NCD). NCDs are among the most common diseases and include obesity, type 2 diabetes mellitus, liver steatosis, cardiovascular diseases, as well as neurodegenerative disorders, and certain types of cancer [1,2]. The World Health Organization now recognizes obesity as a global pandemic. Worldwide, the number of obese children and adults has tripled between 1975 and 2016 [2]. In 2025, about 30% of the US population will have type 2 diabetes, and 50% will be obese [3].

Several studies have shown that the permanent consumption of unhealthy convenience products adversely impacts microbial resilience in the intestine. The intestinal tract is a complex system requiring continuous barrier integrity and regulatory mechanisms to maintain host–microbe interactions and, consequently, immune-tissue homeostasis [4]. The human microbiome shows a high biodiversity in the healthy state and is mainly characterized by the following microbiota: *Bacteroidetes* (*Prevotella*), *Firmicutes* (*Ruminococcus*, *Roseburia*), *Actinobacteria*, *Proteobacteria* and *Verrucomicrobiota* [5]. Symbiotic and commensal microorganisms dominate over opportunistic pathobionts. They are responsible for healthy mucus production, the maintenance of barrier integrity and immune tolerance.

However, the permanent consumption of unhealthy high-calorie convenience products is associated with a reduction in microbial species diversity, an overgrowth of pathobionts, a disturbed barrier integrity and the disruption of the microbial immune homeostasis [6]. Overall, the microbiome represents the interface between host’s nutritional and inflammatory environment. Dietary dysbiosis leads to the release of inflammatory mediators and pathogenic microbial metabolites into the bloodstream [6]. These factors adversely affect systemic metabolism (diet-induced metabolic endotoxemia) and physiological functions of many organs, including intestine, liver, pancreas, bone marrow, spleen and brain [7,8,9,10]. Long-term, disturbed immuno-metabolic processes lead to disturbances in circulating levels of specific lipid and amino acid classes and cholesterol and trigger inflammatory processes in NCDs [11]. We have recently shown in a mouse model of atherosclerosis that feeding a high fat and high cholesterol Western diet (HFD) triggers long-term (epi-)genetic innate immune cell reprogramming, a process known as “trained immunity” [12,13,14]. Four weeks of HFD feeding induced systemic inflammation that subsided after shifting mice to control chow diet (CD). Additionally, HFD triggered a proliferative hematopoietic cell expansion associated with functionally reprogrammed myeloid precursors in the bone marrow compartment. Of note, these responses were maintained over prolonged times even when reversing the diet from HFD to CD, indicating that nutritional composition can induce cellular reprogramming [14]. Questions such as triggers (microbial metabolic products), the duration, the specificity and reversibility of innate immune training in the context of diet feeding still have to be answered. Moreover, it is currently unknown if changes in systemic metabolism due to dietary interventions are long-lasting and how they likely affect acute inflammatory processes, as well as sustained cellular reprogramming.

Recent studies have accentuated the advances in high-throughput metabolomics analysis to unravel the complexity and specificity of metabolic alterations in mouse and human blood and tissues upon dietary intervention [15,16]. Indeed, metabolomics approaches are valuable in identifying diet-related metabolic signatures, which reflect microbial changes, health status and pathophysiological conditions [17,18,19,20]. In a recent paper, Cirulli and co-workers used non-targeted metabolomics and whole-genome sequencing to identify metabolic and genetic signatures in diet-induced obesity. They assessed that the metabolome captures clinically relevant obesity phenotypes and is a better health predictor than genetic risk [21]. Piening and colleagues recently performed a controlled longitudinal weight perturbation study combining multiple omics strategies during weight gain and weight loss in humans. Overall, they observed strong omics signatures upon weight gain (inflammatory responses, insulin resistance), which were partly reversed upon weight loss [22]. Overall, studies that monitor the long-term effects of perturbances in metabolic homeostasis are rare.

Here, we examined systemic metabolic alterations in response to diet and lipopolysaccharide (LPS) challenge. We used high-throughput metabolomics analysis to profile metabolome levels in sera collected from *Ldlr^−/−^* mice fed either CD or HFD and after additional low-dose LPS challenge before sacrifice. HFD feeding, as well as LPS treatment by themselves, triggered pronounced metabolic changes. HFD qualitatively altered the systemic metabolic response to LPS, particularly dynamics for fatty acids and fatty acid-derived bioactive molecules. To identify unique long-term systemic metabolic signatures, mice were fed HFD for four weeks and subsequently reset to CD (HFD > CD). Serum metabolites, being elevated upon HFD feeding, returned to baseline levels analyzed in CD-fed mice. Similarly, metabolic profiling in CD_LPS- and HFD > CD_LPS-fed mice resembled each other.

Overall, we were able to detect profound systemic metabolic rewiring in response to diet and LPS challenge that might be involved in long-term pathophysiological processes and immune cell reprogramming.

## 2. Results

### 2.1. LPS Exposure and HFD Feeding Elicit Distinct and Mutual Changes on Systemic Metabolism

This study’s overall goal was to determine whether *Ldlr^−/−^* mice respond differently to LPS exposure when maintained on CD or HFD, respectively. Additionally, we aimed to study long-term diet effects on systemic metabolism. Therefore, mice were fed either CD and HFD, respectively, for four weeks or four weeks HFD followed by four weeks CD (analysis of long-term diet effects). At the end of dietary intervention, mice received an intravenous low-dose LPS injection (Figure 1a). Sera were collected and subjected to non-targeted unbiased metabolite profiling. A total number of 579 compounds of known identity were detected (Appendix A). Pearson correlation analysis revealed that samples segregated into two major groups based on dietary intervention regimen: Cluster one comprised all HFD samples, and cluster two all CD and HFD > CD samples. Within the different diet treatment groups (CD/HFD > CD), samples separated based on LPS treatment (Figure 1b). Hierarchical clustering analysis confirmed the pronounced effect elicited by HFD feeding. As shown by correlation analysis, CD and HFD > CD samples were highly similar. While these samples also clustered upon additional LPS treatment, HFD samples did less so (Figure 1c). Metabolite-wise hierarchical clustering segregated the dataset into two major clusters, of which cluster one comprised metabolites that were most abundant in the CD and HFD > CD groups, while the second cluster (2–5) comprised metabolites particularly abundant in the HFD and LPS-treated groups. The latter further segregated into sub-clusters with metabolites that were specifically increased upon LPS (3), HFD (4), and both LPS and HFD (5) treatment (Figure 1c). These results indicate that HFD feeding and LPS treatment elicit distinct and mutual changes in the serum metabolome. We next aimed to identify metabolites that were significantly altered between the different treatment groups. We identified 112 significantly altered metabolites for the CD_LPS group, 177 for the HFD and 184 for the HFD_LPS group, compared to the CD group (Figure 1d). In accordance with the correlation and hierarchical clustering analysis, the largest overlap amongst the former three groups was observed between the HFD and HFD_LPS treatment groups (Figure 1d). A considerably smaller overlap was shown between the CD_LPS and HFD, and the CD_LPS and HFD_LPS groups, respectively (Figure 1d).

### 2.2. LPS Challenge Specifically Increases Systemic Protein and Amino Acid Catabolism

We next analyzed changes in the LPS- and HFD-challenged metabolomes in further detail. The CD_LPS group exhibited 51 upregulated and 61 downregulated metabolites compared to the CD-feeding group (Figure 2a,b). *N*-acetyl *N*-Acetyl-1-methylhistidine, 1-docosahexaenoylglycerol, C-glycosyltryptophan and *N*-acetyl-aspartyl-glutamate were among the most significantly altered metabolites in the CD_LPS treatment group (Figure 2c). The induction of lipolysis has been well described in the context of sepsis [23]. Indeed, almost 50% of LPS-modified serum metabolites belonged to lipid classes (Figure 2b). In addition to 1-docosahexaenoylglycerol, a number of other monoacylglycerol and lysolipids were upregulated in the CD_LPS treatment group (Appendix A), and so were several fatty acids (Figure 2d). Mobilization of fatty acids and related lipid metabolites seemed to be limited to distinct fatty acid classes. LPS almost exclusively increased abundances of polyunsaturated fatty acids such as ω-3 and ω-6 fatty acids. With the exception of myristoleate, no mono- or unsaturated fatty acids were mobilized in response to LPS challenge (Figure 2d). Furthermore, we identified several peptides and compounds related to amino acid metabolism being up- or downregulated upon LPS challenge. This is in accordance with literature demonstrating that sepsis triggers changes in systemic protein metabolism [23]. LPS led to a robust decline of a number of amino acids: Ornithine and citrulline, metabolites of the urea cycle, which are known to eliminate amino acid-derived nitrogen in the liver, were downregulated, while the end product urea was increased in the LPS-challenged group (Figure 2d). Guanidinoacetate, being synthesized in the kidney and serving together with ornithine as precursor for the creatine biosynthesis, was decreased, while creatine was increased upon LPS challenge (Figure 2d). Nucleotides were also prominently represented among the upregulated metabolites, and so were nicotinamide and metabolites thereof (Figure 2d). Overall, these results suggest that LPS challenge in CD-fed *Ldlr^−/−^* mice increases systemic protein and amino acid catabolism.

### 2.3. HFD Feeding Is Strongly Impacting Lipid Metabolites

HFD feeding altered a total of 177 biochemicals, of which 95 were up- and 82 were downregulated. (Figure 3a,b). Upon HFD feeding, a high number of lipids and lipid metabolites were significantly altered, particularly in the upregulated fraction. Cholesterol was highly enriched, so was mead acid, a ω-9 fatty acid that is produced from oleic acid in response to deprivation of essential fatty acids [24]. Additionally, sphingomyelin (SM) (d18:2/14:0, d18:1/14:1) and further sphingolipids were highly enriched upon HFD feeding (Appendix A). Of note, linoleate (18:2) and phosphatidylcholine (PC) 18:2/18:3 were among the most significantly downregulated molecules. When analyzing all differentially regulated fatty acids, we observed that, with the exception of mead acid and ω-6 docosapentaenoate (22:5n6), only saturated and monounsaturated fatty acids increased upon HFD feeding. In contrast, several essential polyunsaturated fatty acids were markedly decreased. In particular, ω-3 fatty acid species were strongly affected (Figure 3d). Similarly, we found several phospholipid and plasminogen species, encompassing linoleate (18:2), being downregulated, while a higher number of palmitate (16:0), stearate (18:0) and oleate (18:1)-containing lipids was elevated in the HFD feeding group. We observed a similar regulation pattern for other lipid metabolites, such as monoacylglycerols and lysolipids (Appendix A). Besides lipid-related molecules, numerous amino acid metabolites and diet- and microbiome-derived xenobiotics were decreased upon HFD feeding (Figure 3b). The most strongly downregulated amino acid metabolite was indoleproprionate, a tryptophan derivate that is produced by the gut microbiota [25], supporting intestinal barrier integrity [26]. Likewise, several other microbiome-derived circulating metabolites, important in maintaining the microbiome-immune homeostasis, were decreased upon HFD feeding. Moreover, and just like LPS challenge, HFD feeding altered levels of proteinogenic amino acids and related compounds. In particular, branched-chain amino acids and related metabolites were altered in mice fed a HFD, while LPS challenge affected primarily global amino acid catabolism (Figure 3d).

### 2.4. Effects of LPS Exposure and HFD Feeding on Systemic Metabolism Are Not of Long-Lasting Nature

Global unbiased analysis of the dataset revealed that a switch of HFD-fed *Ldlr^−/−^* mice back to CD restored their metabolome, meaning that HFD-induced metabolic alterations returned to baseline, resembling the metabolome of CD-fed mice (Figure 1b,c). Indeed, we observed that all metabolites, being up- or downregulated, respectively, in the HFD-fed group, returned to baseline levels described in the CD-fed group, in the HFD > CD group (Figure 4a). We have previously observed a state of enhanced LPS-responsiveness in bone marrow myeloid progenitor cells as well as in tissue-resident macrophages after switching mice from HFD back to CD [14]. To assess whether the same applies to systemic metabolism, we analyzed and compared the LPS-challenged metabolome in CD- and HFD > CD-fed groups. Pearson correlation analysis and hierarchical clustering have demonstrated that HFD > CD feeding did not drastically alter systemic metabolic response upon additional LPS challenge (Figure 1b,c). Likewise, we did not observe significant differences in the abundance of LPS-altered metabolites (either up- or downregulated) in the HFD > CD- compared to the CD-fed group (Figure 4b).

### 2.5. LPS Treatment and HFD Feeding Have Synergistic Effects on Systemic Metabolism

To investigate whether the combined treatment of HFD feeding and LPS challenge altered systemic metabolism, we compared systemic metabolite abundances between mice fed CD or HFD diet and additional LPS challenge. We identified 74 significantly altered metabolites, most of them being lipids (Figure 5a,b). Additionally, amino acid metabolites and xenobiotics were altered (Figure 5b). The most dynamically altered metabolites that were already modified by HFD feeding were the following: mead acid, sphingomyelins, BCAA metabolites, essential fatty acids and metabolites thereof such as 9, 10-DiHOME and 12, 13-DiHOME, which are produced from linoleic acid (Figure 5c,d and Appendix A). Notably, the essential ω-3 fatty acids stearidonate, eicosapentaenoate and linoleate were among the most significantly downregulated molecules in the HFD_LPS group (Figure 5d). LPS treatment did not significantly alter systemic levels of these metabolites, yet tended to increase them.

Mobilization of fatty acids from adipose tissue is a metabolic hallmark of inflammation. Likewise, we also found several fatty acids increased upon LPS challenge (Figure 2d). Here, we analyzed whether these metabolites were also mobilized upon LPS challenge in HFD-fed mice. Dihomo-linoleate (20:2n6), dihomo-linolenate (20:3n3 or n6), docosadienoate (22:2n6) were mobilized to a similar degree in the HFD_LPS-treated group; however, levels of the ω-3 fatty acids docosahexaenoate (22:6n3), docosapentaenoate (22:5n3), docosatrienoate (22:3n3), did not increase upon LPS challenge in HFD-fed mice but were significantly reduced compared to the CD_LPS-treated mice (Figure 6a and Appendix A). Next, we analyzed how additional LPS challenge altered serum concentrations of fatty acids (mostly unsaturated and monounsaturated fatty acids) being elevated upon HFD feeding (Figure 3d). None of the HFD-induced saturated fatty acids further increased upon LPS challenge, though circulating levels of laurate (12:0), mead acid (20:3n9), pentadecanoate (15:0), 10-nonadecenoate (19:1n9) and myristoleate (14:1n5) were significantly elevated compared to levels in the CD_LPS group (Figure 5c and Figure 6b,c and Appendix A). For several metabolites and metabolite classes, synergistic effects between HFD feeding and LPS challenge were observed. For example, stearoryl ethanolamide, an endocannabinoid, as well as the dicarboxillic acid tetradecanedioate, were among the most dynamically altered metabolites when comparing the CD_LPS- and HFD_LPS-treated groups. Moreover, when comparing all significantly altered endocannabinoids between the CD-, CD_LPS-, HFD- and HFD_LPS-treated groups, we found that stearoyl ethanolamide, palmitoyl ethanolamide, *N*-oleoyltaurine oleoyl ethanolamide were markedly elevated in the CD_LPS-treated group (Figure 6d and Appendix A). *N*-stearoyltaurine and *N*-palmitoyltaurine were additionally increased in the HFD_LPS-treated group (Figure 6d and Appendix A). Overall, there was a clear trend for all six metabolites being increased in the HFD_LPS compared to the CD_LPS-treated group. (Appendix A). Elevated acylcarnitine serum levels have been detected in sepsis and obesity and are thought to occur as a result of defects in β-oxidation [27,28]. In our diet model, we also identified numerous acylcarnitine species being elevated in the CD_LPS-, as well as the HFD-treated groups (Figure 6e and Appendix A). Additionally, we identified significantly elevated levels of dicarboxilic acids and corresponding metabolites (Figure 6e and Appendix A). Dicarboxilic acids are products of fatty acid ω-oxidation, occurring when β-oxidation is impaired [29].

## 3. Discussion

Consumption of Western-type diets, as well as inflammatory processes, goes along with profound changes in systemic metabolism, which are long-term associated with the development of non-communicable metabolic diseases [30]. As such, diet-induced disturbances of the immuno-metabolic balance lead to several chronic metabolic disorders, including obesity, type 2 diabetes, atherosclerosis and cardiovascular diseases, certain neurodegenerative disorders, as well as certain types of cancer [11,21]. In the present study, we aimed to investigate how changing from a healthy plant-based balanced diet to a Western-type diet, highly enriched in animal fats and refined sugars, adversely affects global metabolism and systemic metabolic responses to LPS challenge. Particularly, we were interested in analyzing diet-dependent metabolic alterations, potentially affecting inflammatory processes and systemic immune responses in the early phase of atherosclerosis development (4 weeks post-diet feeding). Western diet-related increased circulating LDL cholesterol levels have been linked mechanistically and genetically to inflammatory processes and a heightened risk of developing atherosclerosis [31]. Hence, the high-fat diet chosen in this study did not only contain more fat but also varied in the content of cholesterol and the quality composition of the fat source. The influence of HFD was reflected in global metabolic changes, most widely seen in the altered abundance of free fatty acids and associated metabolites. Beyond this, we analysed LPS-distinct and overlapping global metabolic signatures in *Ldlr^−/−^* mice fed either CD or HFD and compared them to metabolic signatures observed only in HFD-fed mice. Profound systemic metabolic rewiring in response to diet might be involved in long-term immune cell reprogramming. We have previously investigated whether high fat/high cholesterol diet feeding triggers trained immunity in the *Ldlr^−/−^* atherosclerosis mouse model [14]. We demonstrated that HFD feeding induced systemic inflammation, shown by elevated circulating inflammatory cytokines, that subsided after shifting mice to control CD. Of note, HFD feeding induced immune cell reprogramming that was maintained over prolonged times, even after reversing the diet from HFD to CD. To identify the long-lasting nature of HFD feeding on systemic metabolism, we compared metabolic signatures in CD-fed, HFD-fed and HFD > CD-fed mice and upon additional LPS challenge. To our surprise, neither HFD > CD feeding nor HFD > CD_LPS treatment re-displayed HFD-induced metabolic alterations. Levels of circulating metabolites were comparable in HFD > CD- and CD-fed mice.

LPS treatment, the causative agent of sepsis, already evoked strong global metabolic rewiring in CD-fed animals, as described previously [23]. Cytokines and hormones released throughout sepsis induce lipolysis in adipose tissue, while β-oxidation decreases [23,32]. In addition, increased protein catabolism and nitrogen loss, resulting from proteolysis in skeletal muscle and hepatic amino acid metabolism and ureagenesis, have been observed [23,33]. Several studies addressed global changes in the serum metabolome in the context of sepsis [27,34,35]. In our model, we applied a low LPS dose to reflect a state of endotoxemia rather than septic shock. However, we also observed dynamic changes in amino acid and lipid metabolism. In the clinical diagnostics, sepsis-induced lipolysis is routinely assessed by measuring global free fatty acid or glycerol concentrations. In our study, we observed that particularly essential polyunsaturated fatty acids were increased in serum by LPS treatment. We also observed increased levels of *N*-Acetyl-Aspartyl-Glutamate (NAAG) in LPS-treated mice. NAAG’s function as a neurotransmitter has been studied extensively. Recent research uncovered that NAAG serves as a glutamate source for tumor and lymphoma cells [36]. While it is well appreciated that amino acids and free fatty acids released during sepsis may serve as nutrients to immune cells, the role of NAAG or the converting enzyme glutamate carboxypeptidase II has not been studied so far.

As expected, cholesterol was one of the metabolites highly enriched upon HFD feeding in our systemic metabolomics analysis. Cholesterol is a well-known risk factor for atherosclerosis [37,38]. Several studies have shown that cholesterol from animal fats is one of the robust drivers of inflammation in atherosclerosis [7,39,40]. HFD feeding also induced increases in fatty acid phospholipids and phospholipid catabolites (e.g., monoacylglycerols, lysolipids and lysoplasmalogens).

It is well appreciated that Westernized nutrition qualitatively and quantitatively alters the intestinal microbial ecosystem, adversely promoting overgrowth of pathogenic strains and altering microbial metabolism [19,41]. In eubiotic microbiomes, symbiotic and commensal microorganisms dominate over opportunistic pathobionts. They are responsible for inhibiting the production of endotoxins in the gut, as well as for maintaining intestinal epithelium homeostasis. Nutritional dysbiosis leads to impaired barrier integrity, loss of local immune tissue homeostasis and to increased levels of plasma endotoxins [42,43]. Here, we observed dynamic changes in serum concentrations of microbiome-derived molecules. The amino acid metabolite indole-3-propionic acid (IPA), a tryptophan derivate that is produced by the gut microbiota, was markedly reduced [25]. Zhao and colleagues have recently shown that IPA inhibits microbial dysbiosis in rats fed HFD. Moreover, IPA induces the expression of tight junction proteins, such as ZO-1 and Occludin, which are important in maintaining barrier integrity, thus reducing plasma endotoxin levels. Additionally, IPA exhibited anti-inflammatory abilities in reducing levels of inflammatory cytokines and repressing liver inflammation. Overall, IPA’s protective role is associated with control of metabolic and inflammatory pathways within the intestinal and hepatic microenvironment [26].

Several other aromatic amino acid metabolites, amongst others daidzein and other isoflavones, being produced through the phenylpropanoid pathway and important in inhibiting intestinal permeability and systemic immunity [25], were decreased in our diet feeding model. Previous studies have shown that a small number of bacteria, including Clostridium sporogenes, a gut bacterium from the phylum Firmicutes, are mainly responsible for metabolizing phenylalanine and tyrosine to their corresponding propionic acid derivatives [25]. Overall, diet-related dysbiosis is associated with imbalanced microbial metabolism, increased production of intestinal endotoxins, systemic metabolic endotoxemia, and immune and metabolic system-linked diseases [42,43].

Of note, Cao and colleagues have recently examined the contribution of gut microbiota-diet interactions to obesity [44]. Using a diet-induced obesity mouse model, they were able to show that some mice were resistant to HFD-induced obesity. While performing 16S rRNA sequencing, they found several microbial alterations potentially being associated with obesity resistance. Overall, differences in gut microbial composition and function might be linked with differences in metabolism and individual’s resistance to HFD-induced obesity. These results are of importance for future studies when analyzing HFD-induced obesity both in human and mice.

Next to alterations in aromatic amino acids, HFD-fed mice also exhibited prominent alterations in several classes of bioactive molecules that have been shown to affect immune processes. In particular, monosaturated and saturated fatty acids (SFA) were highly abundant as a result of HFD feeding. Fatty acids are an important energy source and are readily taken up and stored as triacylglycerides or metabolized by immune cells upon activation [45]. It has been shown that fatty acids affect immune cell homeostasis and function via their metabolism or engagement with specific membrane receptors. Moreover, the fatty acid composition of membrane lipids influences the fluidity of membranes [46,47,48,49]. Otherwise, lipotoxicity, mitochondrial dysfunction and endoplasmatic reticulum stress, all being observed in the context of metabolic diseases, are induced by saturated fatty acids, which are potent drivers of inflammatory processes [47,50,51,52]. Of note, some of the detrimental effects of saturated fatty acids are mitigated by desaturation or the presence of unsaturated fatty acids [50,51,52], highlighting the importance of balanced dietary fats.

Moreover, HFD-fed mice showed significantly higher levels of several medium-chain fatty acids, which are primarily oxidized in the liver [53] and are only poorly incorporated into cellular lipids by non-adipocyte cells [54]. Yet, not much is known whether and how immune cells metabolize medium-chain fatty acids or which pathways are engaged by exposure to them.

In contrast to an LPS challenge, significant decreases were observed in several polyunsaturated fatty acids (PUFAs), particularly the ω-3 variants in HFD-fed mice. ω-3 fatty acids are potent activators of the G-protein-coupled receptor GPR120 [55] and affect systemic energy balance by triggering the release of gut-derived hormones [55,56]. ω-3 Fatty acids are potent anti-inflammatory molecules that also directly act on immune cells and inhibit TLR signaling in macrophages and decrease the Th17/Treg ratio [57,58]. Indeed, the application of ω-3 fatty acids ameliorates atherosclerosis and metabolic disease progression [57,59].

Hence, upon HFD feeding itself, immune cell subsets are exposed to a vastly different mixture of fatty acids. The synergistic effect of HFD feeding and LPS challenge additionally induces a different global metabolic signature than the LPS challenge in CD-fed mice. Under inflammatory conditions, immune cells employ fatty acids to synthesize lipid mediators such as the proinflammatory prostaglandins and leukotrienes and the anti-inflammatory lipoxins, resolvins, maresins and protectins. Prostaglandins, leukotrienes and lipoxins are all generated from arachidonic acid, a ω-6 fatty acid, while resolvins, maresins and protectins are derived from ω-3 fatty acids [60,61,62]. ω-3 fatty acids and their metabolites additionally exert anti-inflammatory effects by interfering with synthesis of ω-6 fatty acid-derived lipid mediators [62]. Thus, in addition to its proinflammatory character, HFD might also inhibit the resolution of inflammation as a result of a lack of certain crucial fatty acids.

Moreover, HFD-fed, as well LPS-challenged mice, displayed significant increases in several long-chain acylcarnitines species and significant decreases in free carnitine and its metabolic precursor deoxycarnitine. Notable increases were also apparent in several dicarboxylic acid species, oxidized lipid intermediates generated via the ω-oxidation pathway. The increase in these metabolites is consistent with impairments in fatty acid oxidation, or alternatively, overwhelmed β-oxidation. It has been described that during periods of fasting, increased lipolysis and oxidation of fatty acids in mitochondria provides most of the energy needed. Fatty acids enter the cytosol from plasma and are transferred into the mitochondria via the palmitoyl-CoA carnitine transferase II shuttle. This import requires carnitine, which is associated with a decline in plasma carnitine levels [63,64]. In contrast, under obese conditions spillover of acetyl- and acyl-CoA due to overwhelmed β-oxidation is buffered by releasing the respective carnitines into plasma [17]. In the context of sepsis, accumulation of acylcarnitines has been linked to dysfunctional β-oxidation [34]. The superinduction of acylcarnitines and dicarboxylic acids observed in HFD-fed mice upon LPS treatment is consistent with further deterioration of systemic fatty acid oxidation [28].

Furthermore, endocannabinoids were among the most dynamically altered metabolites in the HFD_LPS- compared to the CD_LPS-treated group. Endocannabinoids and endocannabinoidome mediators are derived from long-chain fatty acids, and it is therefore predictable that diets rich in certain fatty acids are able to modulate tissue concentrations of the endocannabinoids. Obesity has been linked to higher endocannabinoid plasma and adipose tissue levels, and altered expression of the cannabinoid receptor 1 (CB_1_R). In adipose tissue, CB_1_R-mediated signaling has been shown to increase lipogenesis and reduce mitochondrial biogenesis [65,66,67,68]. In the liver, HFD-increased endocannabinoid levels and CB_1_R signaling contributed to increased fatty acid production. Genetic and pharmacological blocking of the CB_1_R was shown to protect against the development of obesity, hepatic steatosis and related inflammation [69,70]. Indeed, several lines of evidence support a role for endocannabinoids in modulating obesity-induced inflammation in adipose tissue. A study by the Roche group has demonstrated that inhibiting CB_1_R function attenuates LPS-induced TNF*α* and IL-6 expression in human adipocytes [71].

To sum up, cells in different tissue niches and the body’s organs have a constant supply of nutritional metabolites, which are required to keep cellular metabolism and function, and systemic homeostasis. A disturbed supply of certain metabolites, due to altered eating habits, leads to systemic immuno-metabolic imbalances, which derange cellular signaling on certain levels. Long-term, diet-related perturbations, associated with intestinal dysbiosis and altered immune-metabolic signaling, lead to pathophysiological conditions. In this study, we demonstrated the rapid and robust impact of HFD feeding, as well as LPS challenge, on systemic metabolism. Four weeks of dietary intervention profoundly induced systemic metabolic rewiring, which is reflected in the altered abundance of certain lipoproteins, amino acids, as well as immuno-metabolites. In the long run, it will be of interest to study the effects of metabolic changes on immune cell reprogramming, altered inflammatory responses and associated metabolic disease outcomes.

## 4. Materials and Methods

### 4.1. Mice

The Institutional Animal Care and Use Committees of the University of Massachusetts Medical School approved the experiments performed according to local ethics regulations (IACUC 1945, UMass Medical School, Worcester, MA, USA) and NIH guidelines. *Ldlr^−/−^* were initially purchased from The Jackson Laboratory and kept in house. All mice have been previously backcrossed over ten generations to the *C57Bl6/J* background. For all in vivo animal studies (Hifg fat diet feeding studies, LPS challenge) age (8 weeks of age) and sex-matched female wild type, *Ldlr^−/−^* mice were used with five mice per genotype. During experimental settings, mice had ad libitum access to food and water, and were housed under a 12-h light-dark cycle.

### 4.2. Mouse In Vivo Studies

To induce hyperlipidemia and hypercholesterolemia, female mice were fed a high-fat diet (HFD; Teklad 88137) consisting of 17.3% protein, 21.2% fat (saturated fat 12.8%, monounsaturated fat 5.6%, polyunsaturated fat 1%) and 48.5% carbohydrates for four weeks. Chow diet (Prolab Isopro RMH 30; LabDiet) consisted of 25% protein, 14% fat (ether extract) and 60% carbohydrates. To study long-term diet effects on systemic metabolism, female mice were fed a HFD for four weeks, and subsequently subjected to regular chow diet for additional four weeks (HFD > CD). To study additional effects of LPS priming, mice received an intravenous injection of PBS (vehicle control) or *E. coli*-derived ultrapure LPS (0111:B4; 10 µg/mouse) six hours before sacrifice. Blood was collected via cardiac puncture into ethylene-diamine-tetraacetate (EDTA)-lined tubes and immediately placed on ice. After centrifugation at 300× *g* for 10 min, serum was collected from supernatants. Samples were snap-frozen and kept at −80 °C until analysis.

### 4.3. Metabolomic Analysis

The non-targeted metabolomic analysis of serum samples was performed by Metabolon, Inc. (Durham, NC, USA), on a platform consisting of four independent ultrahigh performance liquid chromatography-tandem mass spectrometry (UPLC-MS/MS) methods. Serum samples were extracted with methanol and analyzed as described. Briefly, a Waters ACQUITY ultra-performance liquid chromatography (UPLC) and a Thermo Scientific Q-Exactive (Waltham, MA, USA) high resolution/accurate mass spectrometer interfaced with a heated electrospray ionization (HESI-II) source and Orbitrap mass analyzer (operated at 35,000 mass resolution were utilized for all methods. Sample extracts were dried, then reconstituted in solvents compatible to each of the four methods. Each reconstitution solvent contained a series of standards at fixed concentrations to ensure injection and chromatographic consistency. One aliquot was analyzed using acidic positive ion conditions and chromatographically optimized for more hydrophilic compounds. In this method, the extract was gradient eluted from a C18 column (Waters UPLC BEH C18–2.1 × 100 mm, 1.7 µm, Milford, MA, USA) using water and methanol, containing 0.05% perfluoropentanoic acid (PFPA) and 0.1% formic acid (FA). Another aliquot was analyzed using acidic positive ion conditions; however, it was chromatographically optimized for more hydrophobic compounds. In this method, the extract was gradient eluted from the same aforementioned C18 column using methanol, acetonitrile, water, 0.05% PFPA, and 0.01% FA and was operated at an overall higher organic content. Another aliquot was analyzed using basic negative ion optimized conditions using a separate dedicated C18 column. The basic extracts were gradient eluted from the column using methanol and water, however with 6.5 mM Ammonium Bicarbonate at pH 8. The fourth aliquot was analyzed via negative ionization following elution from a HILIC column (Waters UPLC BEH Amide 2.1 × 150 mm, 1.7 µm) using a gradient consisting of water and acetonitrile with 10 mM Ammonium Formate, pH 10.8. The MS analysis alternated between MS and data-dependent MS*^n^* scans using dynamic exclusion. The scan range varied slightly between methods, but covered 70–1000 m/z. Raw data were extracted, peak-identified, and QC processed using Metabolon’s hardware and software. Compounds were identified by comparison to library entries of purified standards or recurrent unknown entities comprising retention time/index (RI), mass to charge ratio (*m/z*), and chromatographic data (including MS/MS spectral data). Compound assignments were done based on retention index within a narrow RI window of the proposed identification, accurate mass match to the library +/− 10 ppm, and the MS/MS forward and reverse scores between the experimental data and authentic standards. Metabolite abundances were determined by area-under-the-curve quantification of respective peaks.

### 4.4. Statistical Analysis

Analysis of normalized metabolite abundances was performed using Microsoft Excel, GraphPad Prism6 (San Diego, CA, USA) and MetaboAnalyst 4.0 [72]. Principle component analysis (PCA) was performed using MetaboAnalyst to detect outliers. Outliers were verified using GraphPad Prism, which led to the exclusion of four metabolites. To analyze significantly altered metabolites between groups, multiple *t*-tests were performed, followed by false discovery rate (FDR) adjustment of *p*-values. FDR-adjusted *p*-values (*q*-value) of 0.05 or smaller were considered significant. Pearson was performed using MetaboAnalyst 4.0. Hierarchical clustering was performed using Clustvis [73]. For Volcano plot data representation, normalized metabolite data were log-transformed. Heatmaps were generated using Microsoft Excel and Clustvis. Venn diagrams were generated using InteractiVenn [74].

## Figures and Tables

**Figure 1 metabolites-10-00336-f001:**
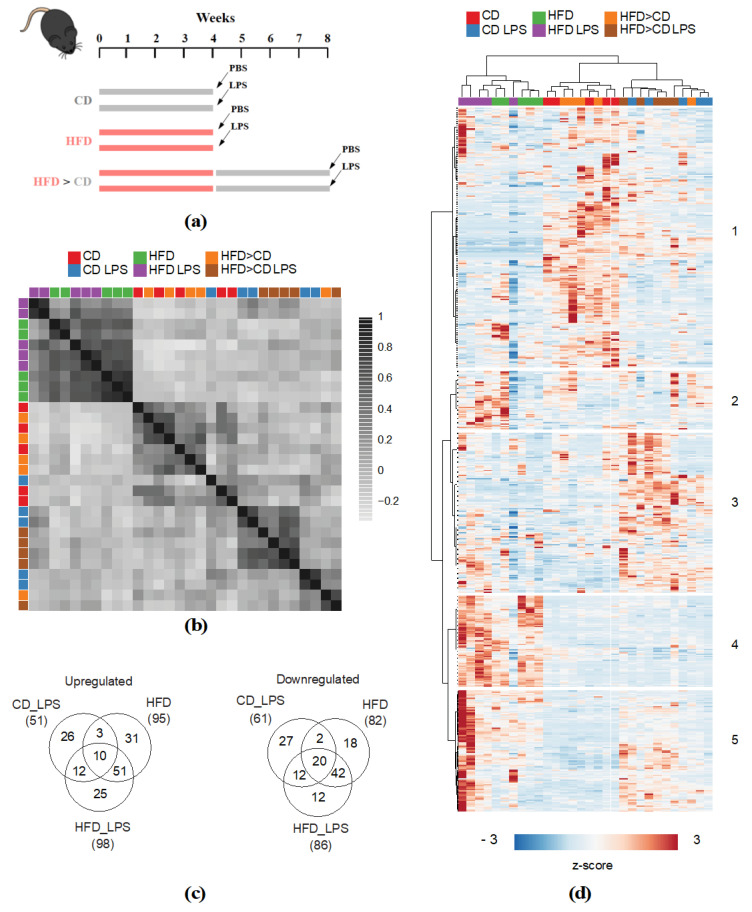
Lipopolysaccharide (LPS) exposure and high-fat-diet (HFD) feeding elicit distinct and mutual changes on systemic metabolism. (**a**) Schematic representation of dietary interventions. *Ldlr*^−/−^ mice were either fed chow diet (CD), HFD for 4 weeks, or HFD for 4 weeks followed by CD for 4 weeks (HFD > CD). After diet intervention, mice received an intravenous LPS injection; (**b**) Pearson correlation analysis of all samples presented in the dataset. Correlation scores were color-scaled; (**c**) Metabolite and sample-wise hierarchical clustering analysis of the entire dataset; (**d**) Venn diagram showing overlap of up- or downregulated metabolites (*q* value < 0.05) between the CD_LPS-, HFD- and HFD_LPS-treated compared to the CD-fed group.

**Figure 2 metabolites-10-00336-f002:**
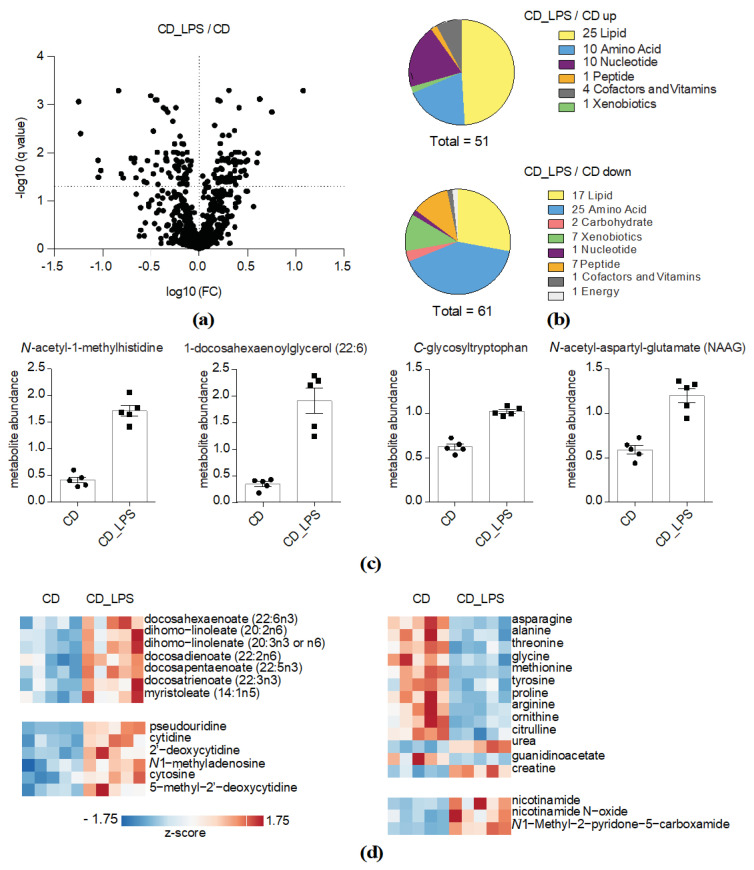
LPS challenge specifically increases systemic protein and amino acid catabolism. (**a**) Volcano plot indicating changes in metabolite abundances between the CD_LPS-treated and the CD-fed group (*q* value < 0.05 is considered significant). −log10 and log10 *q*-values and fold changes, respectively, are depicted; (**b**) Pie charts indicating numbers of altered metabolites in the CD_LPS-treated group compared to the CD-fed group based on their biological classification; (**c**) Abundances of significantly up- or downregulated metabolites in the CD_LPS-treated group. Individual values, means and SEM are presented; (**d**) Representation of significantly up- or downregulated metabolites in the CD_LPS group. Metabolite abundances were *z*-transformed and scaled.

**Figure 3 metabolites-10-00336-f003:**
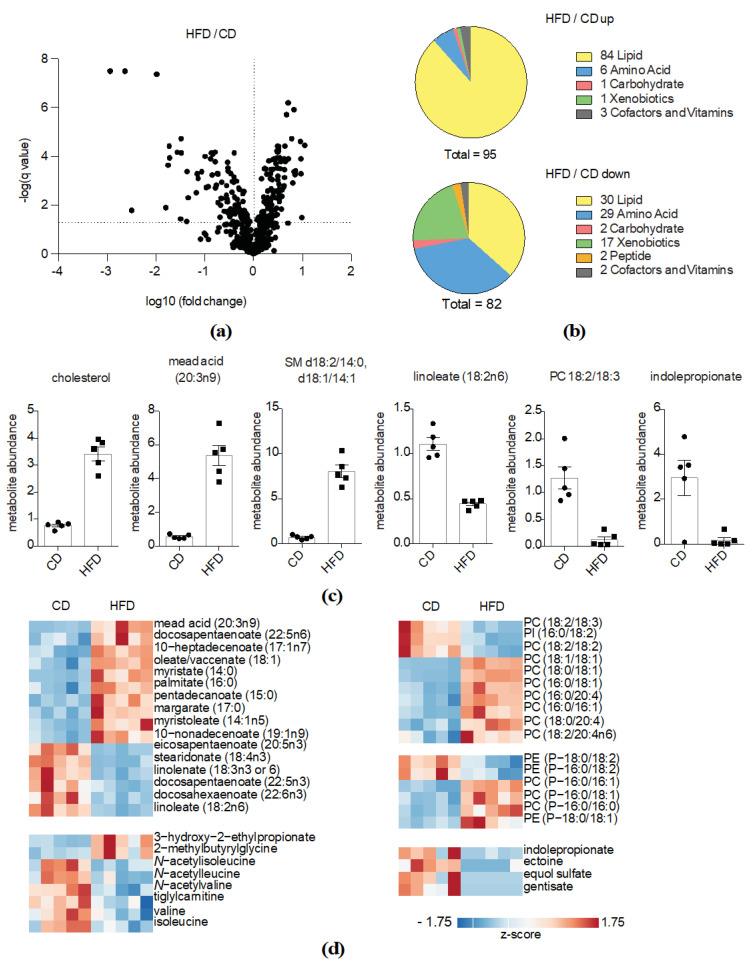
HFD feeding is strongly impacting lipid metabolites. (**a**) Volcano plot indicating changes in metabolite abundances between the HFD- and the CD-fed groups (*q*-value < 0.05 is considered significant). −log10 and log10 *q*-values and fold changes, respectively, are depicted; (**b**) Pie charts indicating numbers of altered metabolites in the HFD-fed group compared to the CD-fed group based on their biological classification; (**c**) Abundances of significantly upregulated metabolites in the HFD-fed group. Individual values, means and SEM are presented; (**d**) Representation of significantly up- or downregulated metabolites in the HFD-fed group. Metabolite abundances were z-transformed and scaled.

**Figure 4 metabolites-10-00336-f004:**
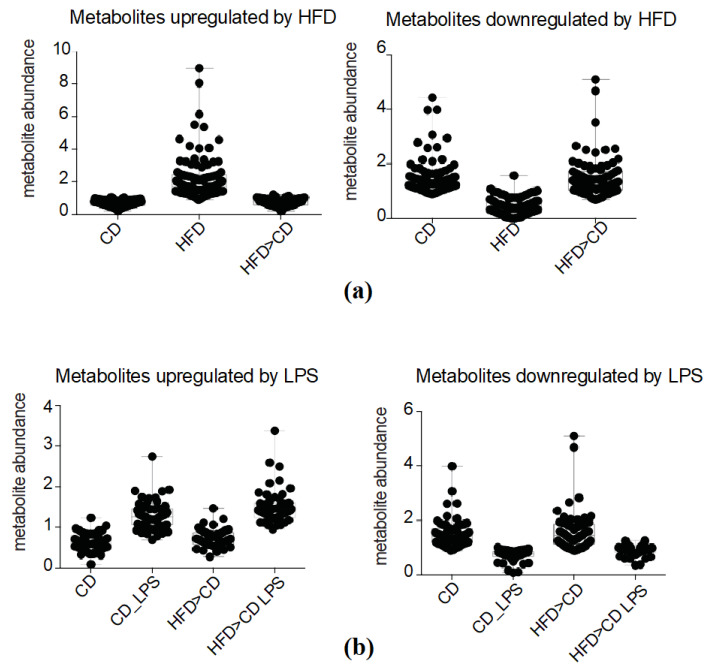
The effects of LPS exposure and HFD feeding on systemic metabolism are not of a long-lasting nature. (**a**) Abundances of metabolites upregulated in the CD-, the HFD- or the HFD > CD-fed groups. Means of individual metabolites are presented; (**b**) Abundances of metabolites upregulated by LPS challenge in the CD-fed group (CD_LPS) and the HFD > CD-fed group (HFD > CD_LPS). Means of individual metabolites are presented.

**Figure 5 metabolites-10-00336-f005:**
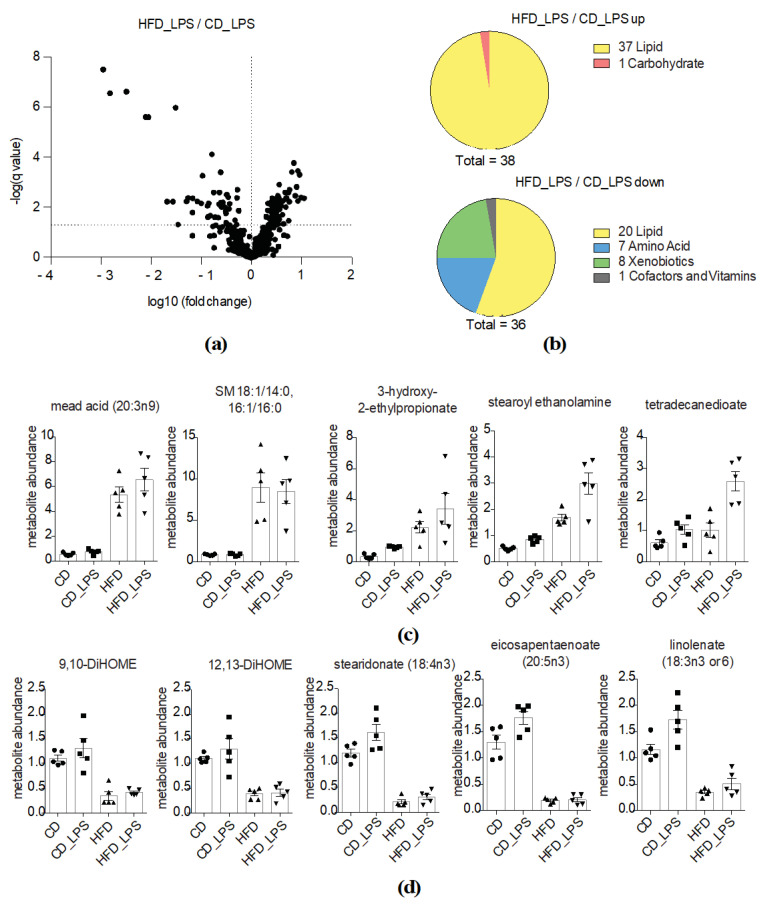
Synergistic effects of LPS treatment and HFD feeding on systemic metabolism. (**a**) Volcano plot indicating changes in metabolite abundances between the HFD_LPS- and the CD_LPS-treated groups (*q*-value < 0.05 is considered significant). –log10 and log10 *q*-values and fold changes, respectively, are depicted; (**b**) Pie charts indicating numbers of altered metabolites in the HFD- compared to the CD-fed group based on their biological classification; (**c**) Abundances of significantly upregulated metabolites in the HFD_LPS-treated group compared to the CD-fed group. Individual values, means and SEM are presented; (**d**) Abundances of significantly downregulated metabolites in the HFD_LPS-treated compared to the CD-fed group. Individual values, means and SEM are presented.

**Figure 6 metabolites-10-00336-f006:**
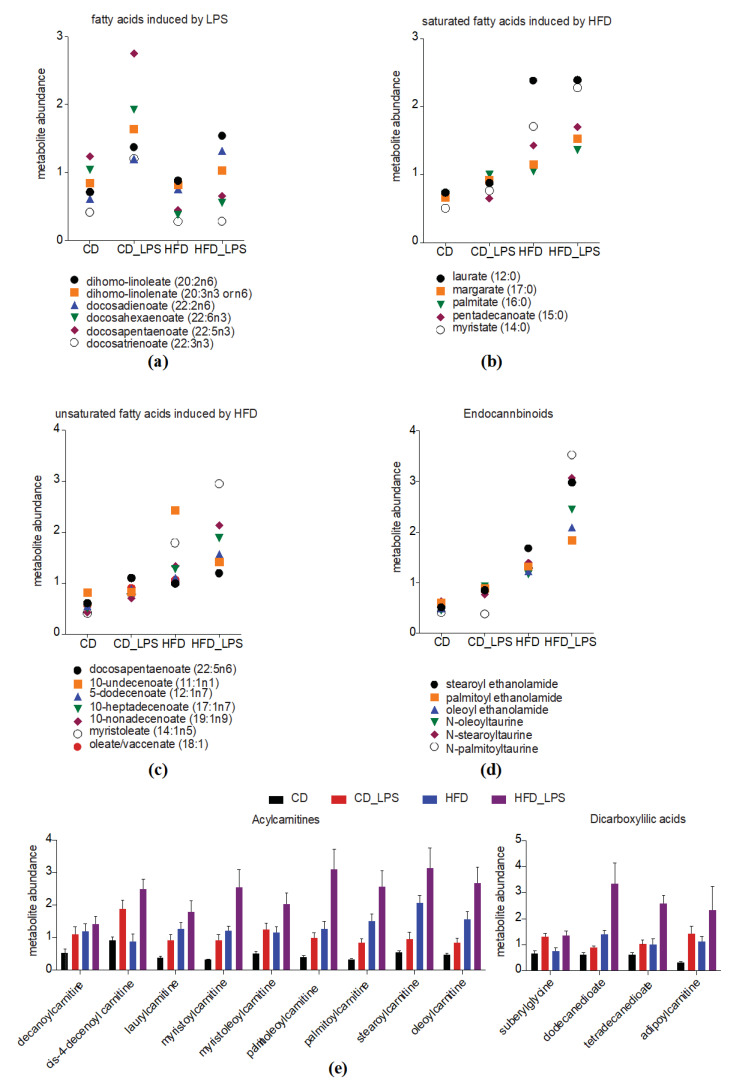
Synergistic effects of LPS treatment and HFD feeding on specific metabolite classes. (**a**–**e**) Abundances of (**a**) Endocannabinoids; (**b**) LPS-upregulated fatty acids; (**c**) HFD-upregulated unsaturated fatty acids; (**d**) HFD-upregulated saturated fatty acids in the CD-, CD_LPS-, HFD- and HFD_LPS-treated groups. Means of individual metabolites are presented; (**e**) Abundances of upregulated acylcarnitines and dicarboxilic acids. Means and SEM are presented.

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
