# Peer review of "Metabolomic Profiling Reveals Distinct and Mutual Effects of Diet and Inflammation in Shaping Systemic Metabolism in Ldlr−/− Mice"

_metabolites, 2020, doi:10.3390/metabo10090336_

Round 1
Reviewer 1 Report
Comments:
Discussion section must include more detailed information regarding gut microbiota (even though not investigated in this study), as high fat diet causes dysbiosis which in turn is associated with lower gut integrity and higher systemic inflammation which affects metabolic pathways. The following part of the result section “The most strongly downregulated amino acid metabolite was indoleproprionate, a tryptophan derivate that is produced by the gut microbiota [25], supporting intestinal barrier integrity [26]. Likewise, several other microbiome-derived circulating metabolites, important in maintaining the microbiome-immune homeostasis, were decreased upon HFD feeding.” should be discussed.
I believe authors should incorporate in the discussion section the results from the following study: DOI: 10.1080/09637486.2019.1686608 , which showed that high fat diet induced obesity caused dysbiosis in the C57BL/6J mice and changed microbiome resistance while affecting metabolic profile.
Introduction In the sentence “… and intestinal barrier integrity.” Authors should add appropriate reference: DOI: 10.1080/09637486.2019.1580682
Author Response
Response to Reviewer 1 Comments
Point 1: Discussion section must include more detailed information regarding gut microbiota (even though not investigated in this study), as high fat diet causes dysbiosis which in turn is associated with lower gut integrity and higher systemic inflammation which affects metabolic pathways. The following part of the result section “The most strongly downregulated amino acid metabolite was indoleproprionate, a tryptophan derivate that is produced by the gut microbiota [25], supporting intestinal barrier integrity [26]. Likewise, several other microbiome-derived circulating metabolites, important in maintaining the microbiome-immune homeostasis, were decreased upon HFD feeding.” should be discussed.
I believe authors should incorporate in the discussion section the results from the following study: DOI: 10.1080/09637486.2019.1686608, which showed that high fat diet induced obesity caused dysbiosis in the C57BL/6J mice and changed microbiome resistance while affecting metabolic profile.
Response 1: We included now more detailed information regarding gut microbiota and diet-related dysbiosis in the discussion section. We particularly focused on the impact of HFD on the presence of certain metabolites, such as indoleproprionate, that are important in maintaining the microbiome-immune homeostasis. We incorporated results from the recommended study (DOI: 10.1080/09637486.2019.1686608) as well.
Changes within the text are marked in red.
Point 2: Introduction: In the sentence “… and intestinal barrier integrity.” Authors should add appropriate reference: DOI: 10.1080/09637486.2019.1580682.
Response 2: We now added the appropriate reference (DOI:10.1080/09637486.2019.1580682) in the introduction section. We additionally included a short paragraph about gut microbiota and diet-related dysbiosis in the introduction section.
Changes within the text are marked in red.
Reviewer 2 Report
The authors present a serum metabolomics study where there assess the effects of a high fat diet and low dose LPS challenge on metabolic response. Furthermore, they illustrate that switching from a high fat diet back to a control diet for a period of time (4 weeks) mitigates the effects a high fat diet has on serum metabolites. In particular they noted that changes in serum fatty acids/fatty acid metabolites.
This is an interesting, and very well written article (I wish all the articles I peer reviewed were this well written!). The article is scientifically sound, there are a few very minor points I have highlighted below, once they are fixed, I would highly recommend it for publication:
ABSTRACT: “mice fed a WD” …do you mean HFD?
Fig 1A. Looking at the diagram, it appears that group 1 (the top 2 lines) were fed a CD for 8 weeks. However in the description it says:
“Ldlr-/- mice were either fed CD, HFD for 4 weeks, or HFD for 4 weeks followed by CD for 4 weeks (HFD>CD).”
I would suggest putting the names of the groups in place of the numerals 1,2,3 in Fig 1A (i.e. calling them CD, HFD and HFD>CD, respectively) to make it consistent and clearer with the way these groups are referred to in rest of the text.
Fig 1D – the “LPS” group….I would recommend renaming “CD_LPS” to make consistent
Page 5:
“C-glycosyltryptophan and N-acetyl-aspartyl-glutamate were among the most significantly altered metabolites in the CD_LPS treatment group (Figure 1C).” - I think this is meant to be Fig 2C?
Fig 4 – were these data z-transformed and scaled? If so please note in the figure legend.
Page 14:
“We observed dynamic changes in serum concentrations of microbiome-derived molecules, indicating that short periods of HFD feeding already result in a profound shift in microbial communities.”
Could you please reference the particular microbiome-derived molecules (e.g. indolepropionate, etc) that you are referring to.
References:
29 - ? correct journal name
37, 38, 43 – journal name: “Nature Publishing Group” – please correct
53 – name error (Christoph Thiele, C. P. D. H. K. K. A. G. M. S. K. P. D. L. J. S. A. S. A. S. A. L. K.) and is journal name missing?
Author Response
Point 1: Abstract: “mice fed a WD” …do you mean HFD?
Response 1: We meant “HFD” and changed it in the text.
Changes are marked in red.
Point 2: Fig 1A: Looking at the diagram, it appears that group 1 (the top 2 lines) were fed a CD for 8 weeks. However, in the description it says: “Ldlr-/- mice were either fed CD, HFD for 4 weeks, or HFD for 4 weeks followed by CD for 4 weeks (HFD>CD).” I would suggest putting the names of the groups in place of the numerals 1,2,3 in Fig 1A (i.e. calling them CD, HFD and HFD>CD, respectively) to make it consistent and clearer with the way these groups are referred to in rest of the text.
Response 2: We thank the reviewer for this suggestion. We adapted Figure 1A, to make it more comprehensible for the reader.
Point 3: Fig 1D – the “LPS” group…. I would recommend renaming “CD_LPS” to make consistent.
Response 3: We thank the reviewer for this suggestion. We changed “LPS” in Figure 1D to “CD_LPS” to make it more consistent.
Point 4: Page 5: “C-glycosyltryptophan and N-acetyl-aspartyl-glutamate were among the most significantly altered metabolites in the CD_LPS treatment group (Figure 1C).” - I think this is meant to be Fig 2C?
Response 4: This is correct. We changed ‘Figure 1C’ to ‘Figure 2C’. Changes are marked in red.
Point 5: Fig 4.: Were these data z-transformed and scaled? If so, please note in the figure legend.
Response 5: These data are relative abundance values as provided in the supplemental table. They are not z-transformed.
Point 6: Page 14: “We observed dynamic changes in serum concentrations of microbiome-derived molecules, indicating that short periods of HFD feeding already result in a profound shift in microbial communities.” Could you please reference the particular microbiome-derived molecules (e.g. indolepropionate, etc) that you are referring to.
Response 6: We adapted this part of the discussion and added the particular microbiome-derived molecules.
Point 7A:
References: Reference 29- correct journal name?
Response 7A: We re-checked, this is the correct journal name.
Point 7B: References 37, 38, 43 – journal name: “Nature Publishing Group” – please correct.
Response 7B: We corrected the journal names. Changes are marked in red.
Point 7C: Reference 53 – name error (Christoph Thiele, C. P. D. H. K. K. A. G. M. S. K. P. D. L. J. S. A. S. A. S. A. L. K.) and is journal name missing?
Response 7C: We corrected the names and added the journal name. Changes are marked in red.